

# Exceptional points in the Baxter-Fendley free parafermion model

**Robert A. Henry**$^\star$ **and Murray T. Batchelor**$^\dagger$

Mathematical Sciences Institute, The Australian National University,
Canberra ACT 2601, Australia

$\star$ robert.henry@anu.edu.au , $\dagger$ murray.batchelor@anu.edu.au

## Abstract

Certain spin chains, such as the quantum Ising chain, have free fermion spectra which can be expressed as the sum of decoupled two-level fermionic systems. *Free parafermions* are a generalisation of this idea to $\mathbb{Z}_N$-symmetric clock models. In 1989 Baxter discovered a non-Hermitian but $\mathcal{PT}$-symmetric model directly generalising the Ising chain, which was later described by Fendley as a free parafermion spectrum. By extending the model's magnetic field parameter to the complex plane, it is shown that a series of exceptional points emerges, where the quasienergies defining the free spectrum become degenerate. An analytic expression for the locations of these points is derived, and various numerical investigations are performed. These exceptional points also exist in the Ising chain with a complex transverse field. Although the model is not in general $\mathcal{PT}$-symmetric at these exceptional points, their proximity can have a profound impact on the model on the $\mathcal{PT}$-symmetric real line. Furthermore, in certain cases an exceptional point may appear on the real line (with negative field).



# 1 Introduction

The free parafermion model is a quantum spin chain where the spin sites are generalised to "clocks" with $N$ symmetric states. On a chain of length $L$ with open boundary conditions, it has the Hamiltonian

$$H = -\sum_{j=1}^{L-1} Z_j^\dagger Z_{j+1} - \lambda \sum_{j=1}^{L} X_j, \tag{1}$$

where $X$ and $Z$ are generalisations of the Pauli matrices given by

$$Z = \begin{pmatrix} 1 & 0 & 0 & \dots & 0 \\ 0 & \omega & 0 & \dots & 0 \\ 0 & 0 & \omega^2 & \dots & 0 \\ \vdots & \vdots & \vdots & \ddots & \vdots \\ 0 & 0 & 0 & \dots & \omega^{N-1} \end{pmatrix}, \qquad X = \begin{pmatrix} 0 & 0 & 0 & \dots & 0 & 1 \\ 1 & 0 & 0 & \dots & 0 & 0 \\ 0 & 1 & 0 & \dots & 0 & 0 \\ \vdots & \vdots & \vdots & \ddots & \vdots & \vdots \\ 0 & 0 & 0 & \dots & 1 & 0 \end{pmatrix}. \tag{2}$$

The parameter $\lambda$ is a constant assumed to be real and positive in the literature, and $\omega = e^{2\pi i/N}$ is an $N$th root of unity.

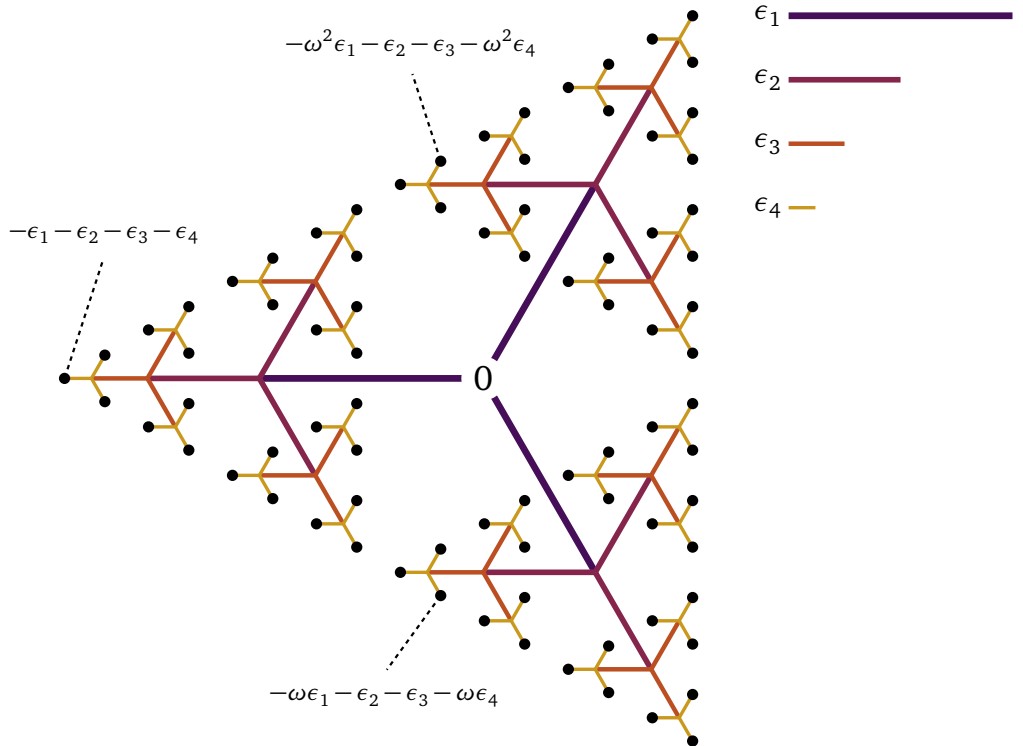

Figure 1: A free parafermion spectrum in the complex plane, for $N = 3$, $L = 4$. The black dots are the $N^L$ energy eigenstates. The spectrum is built up by starting at zero and adding each parafermion multiplied by a power of the root of unity $\omega$, as per Eq. (3). Here the values of $\epsilon_j$ are arbitrary and for most realistic values the paths would overlap each other, but will have the same essential branching structure. Algebraic expressions are shown for some example states, including the ground state $E_0 = -\epsilon_1 - \epsilon_2 - \epsilon_3 - \epsilon_4$.

When $N = 2$, the model reduces to the widely studied transverse field Ising model, with $X$ and $Z$ reducing to the Pauli matrices $\sigma^x$ and $\sigma^z$. For $N > 2$, the model is non-Hermitian and has a complex spectrum. The energy spectrum is known exactly for all $N$, $L$ and $\lambda$, taking the form of *free parafermions*. This form was first derived by Baxter, who also formulated the model [1, 2], and later explored in detail by Fendley using an algebraic approach with parafermions defined by the Fradkin-Kadanoff transformation [3]. Some key elements of Fendley's approach are described in Section 2.

The free parafermion model is related to the classical $\tau_2$ model, in that the transfer matrix of the classical model commutes with the quantum chain's Hamiltonian. The $\tau_2$ model was essential to the solution of the chiral Potts model [4] and has been further explored by Baxter [5], and Au-Yang and Perk [6,7].

In his solution, Fendley developed interesting algebraic techniques which were also applied to multispin free fermion systems [8]. This approach was later adopted and generalised by Alcaraz and Pimenta to a class of multispin free fermion and free parafermion models [9–11].

Free parafermions are a natural generalisation of the concept of free fermions. In a free fermion system, the energy eigenvalues are a sum of a set of $L$ quasienergies, each of which is multiplied by a positive or negative sign, giving $2^L$ combinations which determine the $2^L$ eigenvalues of the Hamiltonian. For free parafermions, instead of a positive or negative contribution, each quasienergy is multiplied by an $N$th root of unity, i.e., a power of $\omega$. Thus a general energy eigenvalue is given by

$$E = -\sum_{j=1}^{L} \omega^{q_j} \epsilon_j \,, \tag{3}$$

where $\epsilon_j$ are the quasienergies, and $q_j \in \{0,\ldots, N-1\}$. Each possible combination of $q_j$ values determines a different eigenvalue, giving all $N^L$ states. Figure 1 provides an illustration of such a spectrum.

The quasienergies $\epsilon_j$ are real and positive for real and positive $\lambda$. They were determined in Baxter's original paper, and in a number of other ways in the literature. Alcaraz et al. [12] calculate them in terms of a quasimomentum variable $k$:

$$\epsilon_{k_j} = (1 + \lambda^N + 2\lambda^{N/2} \cos k_j)^{1/N} \,, \tag{4}$$

where $k_j$ are the $L$ solutions in the interval $(0, \pi)$ of

$$\sin([L+1]k) = -\lambda^{-N/2} \sin(Lk) \,. \tag{5}$$

This formulation is convenient for the analytic approach used in this work. The quasienergies are equivalently given by the eigenvalues of matrix [13]

$$\mathcal{M} = \begin{pmatrix} 1 & g & & & \\ g & 1+g^2 & g & & \\ & g & 1+g^2 & \ddots & \\ & & \ddots & \ddots & g \\ & & & g & 1+g^2 \end{pmatrix}, \tag{6}$$

where $g = \lambda^{-N/2}$. The quasienergies are then $\epsilon_j = \lambda a_j^{1/N}$, where $a_j$ is an eigenvalue of $\mathcal{M}$. Diagonalising $\mathcal{M}$ is an efficient way of finding the quasienergies numerically, and is used to obtain most of the numerical results found in this paper. Alcaraz et al. [12] also determined other quantities including the critical heat exponent, and the ground state energy in the thermodynamic limit:

$$e_\infty(\lambda) = -\,_2F_1\left(-\frac{1}{N}, -\frac{1}{N}; 1; \lambda^N\right), \tag{7}$$

where $_2F_1$ is the hypergeometric function.

In Baxter's and Fendley's work, more general models with arbitrary real and positive coefficients on each term in Eq. (1) are considered. These have the effect of changing the quasienergies but do not change the free parafermion character of the spectrum, and cannot lead to the simple closed forms shown above. Only uniform models are considered in this work.

## 1.1 Periodic systems and non-Hermitian physics

The free parafermion model exhibits substantially different behaviour under periodic boundary conditions, which are implemented by adding the boundary term $-Z_L^\dagger Z_1$ to Eq. (1). In fact, the free parafermion solution no longer applies, unlike many free fermion systems where the system breaks into free fermionic momentum sectors. Surprisingly, the energy of the system depends on the boundary conditions even in the thermodynamic limit ($L \to \infty$), which is impossible in a Hermitian system [13]. This is an example of a *non-Hermitian skin effect* and has recently been observed in a variety of non-Hermitian systems [14–16].

Recently, there has been extensive activity surrounding non-Hermitian models. In many cases they have interesting behaviour relating to *exceptional points* (EPs), which are isolated points in the parameter space where the Jordan structure of the Hamiltonian changes. In the physics literature, EPs refer to a non-trivial block forming, i.e., an off-diagonal element in the normal form. This is associated with degenerate eigenvalues, with the corresponding right eigenvectors becoming parallel, and the corresponding left eigenvectors becoming orthogonal to the right eigenvectors. This is only possible for non-Hermitian matrices and is leads to a variety of novel physics, including the non-Hermitian skin effect, generalised geometric phases, the anomalous bulk-boundary correspondence, and exotic phase transitions. See Bergholtz et al. [17] for a recent review of these developments, and Ashida et al. [18] for an excellent extensive review of non-Hermitian physics in general, including thorough introductory material on Jordan block structure and EPs.

## 1.2 Symmetries

The free parafermion model has a $\mathbb{Z}_N$ symmetry generated by the operator $\prod_{j=1}^L X_j$, which rotates each clock by one place. It also has parity-time ($\mathcal{PT}$) symmetry, with the action of the operators $\mathcal{P}$ and $\mathcal{T}$ being as follows:

$$\mathcal{P} Z_j \mathcal{P} = Z_{L+1-j}, \qquad \mathcal{P} X_j \mathcal{P} = X_{L+1-j}, \tag{8}$$

$$\mathcal{T} Z_j \mathcal{T} = Z_j^\dagger, \qquad \mathcal{T} X_j \mathcal{T} = X_j. \tag{9}$$

The parity operator $\mathcal{P}$ inverts or flips the lattice, and the time-reversal operator $\mathcal{T}$ conjugates all numbers, including $\lambda$. There has been great interest in $\mathcal{PT}$-symmetric non-Hermitian systems, beginning with the work of Bender and Boettcher [19]. These systems, while not Hermitian, have real energy spectra when the $\mathcal{PT}$ symmetry is unbroken, while $\mathcal{PT}$-broken energy states appear in complex conjugates. With an appropriate metric they have unitary time evolution [20, 21], even if the symmetry is broken, and thus most of the physics of a standard closed quantum system still applies to them. If the $\mathcal{PT}$ symmetry is broken, some eigenvalues appear in conjugate pairs. In the past two decades, non-Hermitian and particularly $\mathcal{PT}$-symmetric physics has been applied in a large variety of novel experiments and theoretical work. Many examples of this are covered in the review of Ashida et al. [18].

## 2 Fendley's solution

Fendley's solution [3] expresses the Hamiltonian in terms of parafermions given by the Fradkin-Kadanoff transformation:

$$\psi_{2j-1} = \left( \prod_{k=1}^{j-1} X_k \right) Z_j \, ,$$

$$\psi_{2j} = \omega^{(N-1)/2} \psi_{2j-1} X_j \, . \tag{10}$$

The $2L$ parafermions $\psi_j$ satisfy a generalised Clifford algebra, which reduces to Majorana fermions for $N = 2$:

$$(\psi_a)^N = 1, \; \psi_a^\dagger = (\psi_a)^{N-1}, \qquad \psi_a \psi_b = \omega \psi_b \psi_a \, (a < b) \, . \tag{11}$$

The Hamiltonian can then be rewritten as

$$H = \omega^{(N-1)/2} \sum_{a=1}^{2L-1} t_a \psi_{a+1} \psi_a^\dagger \, , \tag{12}$$

where $t_a = 1$ ($a$ odd), $t_a = \lambda$ ($a$ even) for the uniform case. Fendley re-expresses the Hamiltonian in various forms by using the Clifford algebra. For our purposes one of these forms is sufficient, which makes the generalisation of free fermions clear. The Hamiltonian is expressed as a sum of decoupled $N$-level systems:

$$H = - \sum_{k=1}^{L} \Xi_k \, , \tag{13}$$

where $[\Xi_k, \Xi_{k'} = 0]$, and each $\Xi_k$ has $N$ distinct eigenvalues given by $\epsilon_k, \omega \epsilon_k, \ldots, \omega^{N-1} \epsilon_k$, where $\epsilon_k$ are the quasienergies defining the spectrum in Eq. (3). This is a simplified version of a more general form given by Fendley which also covers a series of higher Hamiltonians, and is not necessary for the analysis given in this work. Since the $\Xi_k$ commute, states can be chosen that are simultaneous eigenstates of each $\Xi_k$, and thus a state with any energy of the form of Eq. (3) may be selected.

The operators $\Xi_k$ only depend on $\lambda$ through the quasienergies $\epsilon_k$ and through a linear transformation relating them to the basic parafermion operators $\psi_j$, which is common to all the $\Xi_k$. This means that if two $\epsilon_k$ were to become degenerate, so would the corresponding $\Xi_k$ and therefore many of the eigenvectors of $H$, producing an exceptional point. Such a degeneracy does not occur for the positive real couplings considered in the literature. For these values, the model has $N^L$ distinct eigenstates and is always diagonalisable, despite being non-Hermitian. The essential result of this paper is that such quasienergy degeneracies *can* occur for isolated complex values of $\lambda$, producing EPs of the full Hamiltonian.

## 3 Motivation from real $\lambda$

In our previous work on correlations in the free parafermion model [22], it was noted that end-to-end correlation functions of the form $\langle\langle Z_1^\dagger Z_L \rangle\rangle$ diverge as the system size increases. This behaviour is characteristic of the presence of an EP, but can be examined more directly by considering the *ground state fidelity*. Fidelity has various definitions in the literature, always relating to the overlap of two quantum states with different parameters. Recently, Tzeng et al. [23,24] have used a non-Hermitian fidelity to explore EPs, defined by

$$\mathcal{F}(\lambda) = \langle L(\lambda) | R(\lambda + \delta) \rangle \langle L(\lambda + \delta) | R(\lambda) \rangle \, , \tag{14}$$

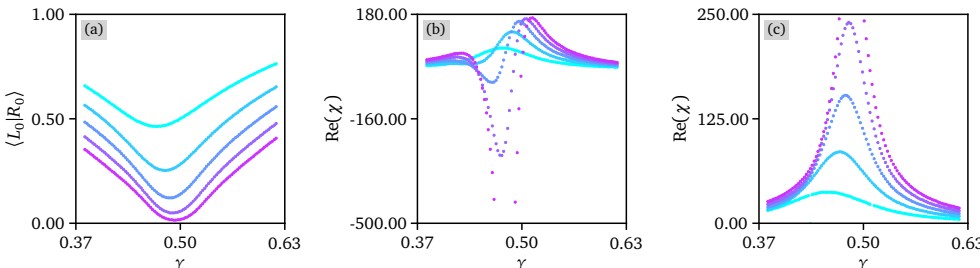

Figure 2: Overlap and fidelity susceptibility for $L = 10$ (cyan), 15, 20, 25, 30 (purple). (a) Left-right ground state overlap for $N = 3$. (b) Fidelity susceptibility for $N = 3$. (c) Fidelity susceptibility for $N = 2$. $\gamma$ is a rescaled analog of $\lambda$, with $\lambda = \frac{\gamma}{1-\gamma}$. The critical point is at $\gamma = 0.5$.

where $\delta$ is a small parameter, $\langle L|$ and $|R\rangle$ are the left and right ground states (or any state of interest), and $\lambda$ could more generally be replaced by any parameter of the Hamiltonian. The fidelity susceptibility $\chi$ is the second-order expansion coefficient in $\delta$, which is approximately

$$\chi \approx \frac{1-\mathcal{F}}{\delta^2}. \tag{15}$$

The first order coefficient vanishes, i.e., $\mathcal{F} = 1 - \chi\delta^2 + \mathcal{O}(\delta^3)$. The key result of Tzeng et al. is that at a critical point, $\mathrm{Re}(\chi) \to +\infty$, while at an EP $\mathrm{Re}(\chi) \to -\infty$.

Figure 2 shows fidelity susceptibility data for the free parafermion model obtained using the density matrix renormalisation group method, with $\delta = 10^{-5}$. The $N = 2$ Hermitian Ising case (c) shows the characteristic behaviour of a critical point: as $L$ increases, $\mathrm{Re}\,\chi \to \infty$. The critical point is not reached for any finite $L$ as the gap does not close, so $\mathrm{Re}\,\chi$ remains finite. For $N = 3$, $\mathrm{Re}\,\chi \to -\infty$ instead, and the overlap of the left and right states approaches zero. This is characteristic of the presence of an EP. However, while $\mathrm{Re}\,\chi$ grows rapidly, it remains finite, indicating that the system is not precisely at the EP for any value of $\lambda$. Unlike a critical point which exists only in the limit $L \to \infty$, EPs typically exist even for finite $L$. This lead us to the idea that an EP could be somewhere in the complex $\lambda$ plane such that it approaches the critical point as $L \to \infty$.

## 4   Complex $\lambda$ and exceptional points

In previous works, the parameter $\lambda$ is assumed to be real and positive. Many existing results, including the free parafermion spectrum, extend directly to any complex value of $\lambda$. This has interesting consequences: the quasienergies $\epsilon_j$ also become complex, and it becomes possible for two of these quasienergies to coincide, producing an EP. The analysis of Baxter, Fendley, and Alcaraz et al. [12] which leads to Eqs. (5) and (7) remains valid for complex $\lambda$. Figure 3 provides numerical confirmation of this, and demonstrates the basic effect of complex $\lambda$ on the spectrum and quasienergies. Complex values of $\lambda$ are parametrised with an angle $\phi$ as follows:

$$\lambda = |\lambda|e^{2\pi i\phi/N}. \tag{16}$$

Due to the model's $\mathbb{Z}_N$ symmetry, a rotation of $\omega$ has the same spectrum as zero rotation, corresponding to $\phi = 1$. This rotation still permutes the states. More generally, a rotation with $\phi > 1$ has the same spectrum as $\phi \pmod 1$. All possible spectra can be seen in the interval $\phi \in [0, 1)$.

Figure 3: Energy spectra (left) and quasienergies (right) for the free parafermion model with $N = 3$, $L = 4$, $\lambda = 1$ and various values of $\phi$. The energies (left) show values obtained from the quasienergies (blue dots), and values obtained from exact diagonalisation of the full Hamiltonian (red crosses).

The $\mathbb{Z}_N$ symmetry also implies that any particular quasienergy could be multiplied by $\omega$ to produce an identical spectrum. This is also reflected in the fact that Eq. (4) has $N$ solutions for a given $k_j$. In the real positive $\lambda$ case the quasienergies are taken as real, but could equivalently be be proportional to powers of $\omega$. For complex $\lambda$, we generalise this convention by taking the quasienergies to have arguments in the interval $(-\pi/N, \pi/N]$, i.e., the choice with the smallest complex argument is taken.

## 4.1 Complex rotations and $\mathcal{PT}$ antisymmetry

Throughout this paper, complex values of $\lambda$ are introduced directly into Eq. (1). However, the complex rotation could also be applied to the $Z$ term of $H$, or partially to each term. The difference between these choices is an overall rotation in the complex plane, which has no effect on the location of the EPs and other physics discussed in this work.

For most values of $\phi$, $H$ is no longer $\mathcal{PT}$-symmetric, which is reflected in the spectrum losing its reflection symmetry across the real axis, so the eigenvalues no longer appear as conjugate pairs (or real). The case $\phi = 0.5$ is special, as the quasienergies appear in conjugate pairs and the symmetry of the spectrum is restored. This can be seen in Figure 3 (bottom row). Interestingly, this is not $\mathcal{PT}$ symmetry, as applying $\mathcal{PT}$ interchanges the conjugate pairs of quasienergies, but is symmetric in $(\mathcal{PT})^2$, which may be interpreted as $\mathcal{PT}$ antisymmetry. This special case is not the focus of this work but may warrant further investigation.

## 4.2 Quasienergy degeneracies

For real positive $\lambda$, the quasienergies $\epsilon_j$ are always positive and distinct. For complex $\lambda$, the quasienergies are complex, and a pair of them may become equal at certain values of $\lambda$, which depend on $L$ and $N$. This was initially observed using numerical data such as can been seen in Fig. 4. As described in Section 2, such quasienergy degeneracies necessarily lead to exceptional points of the full Hamiltonian. As such, we will from now on refer to quasienergy degeneracies simply as EPs.

The locations of the EPs can be determined analytically by finding repeated roots of Eq. (5), meaning that both it and its derivative are satisfied:

$$\sin([L+1]k) + \lambda^{-N/2}\sin(Lk) = 0,\tag{17}$$

and

$$(L+1)\cos([L+1]k) + L\lambda^{-N/2}\cos(Lk) = 0.\tag{18}$$

The EPs occur at pairs of values $k_{EP}$ and $\lambda_{EP}$ which satisfy these equations simultaneously. In other words, the EPs occur only at particular values of $\lambda$, and at those values they appear as a repeated root $k_{EP}$ of Eq. (5), which gives two degenerate quasienergies. Eliminating $\lambda_{EP}$ determines $k_{EP}$ as the solution to

$$(L+1)\sin(Lk_{EP})\cos([L+1]k_{EP}) - L\sin([L+1]k_{EP})\cos(Lk_{EP}) = 0,\tag{19}$$

which may be simplified somewhat to

$$\sin([2L+1]k_{EP}) - (2L+1)\sin(k_{EP}) = 0,\tag{20}$$

with the corresponding value of $\lambda_{EP}$ given by

$$\lambda^N = \left[\frac{-\sin([L+1]k_{EP})}{\sin(Lk_{EP})}\right]^2.\tag{21}$$

In fact, all $N$ complex values of $\lambda$ that solve Eq. (21) for a given $k_{EP}$ are EPs. These different values occur at rotations of $\exp(2\pi i/N)$ relative to each other, and have identical spectra. Figures 4 and 5 demonstrate how solutions of Eq. (19) appear in the complex plane, and how they correspond to quasienergy degeneracies (and hence EPs). The four complex quadrants of $k$ produce identical values of $\lambda_{EP}$ (and identical quasienergies), so only one quadrant needs to be considered. There are in general $L-1$ solutions to Eq. (19), giving $N(L-1)$ EPs. The figures show the relative difference between the smallest quasienergy and the second smallest, defined as

$$\Delta\epsilon_{12} = \min_{j=2}^{L}\frac{|\epsilon_1 - \epsilon_j|}{|\epsilon_1 + \epsilon_j|}, \tag{22}$$

where $\epsilon_1$ is the smallest quasienergy in absolute value. An obvious alternative would be the smallest difference between *any* two quasienergies. The above $\Delta\epsilon_{12}$ is used because the EPs are observed numerically to always occur between the two smallest magnitude quasienergies. Taking the smallest difference between any two quasienergies gives the same roots in the complex plane, but makes their appearance much less distinct (in, e.g., Figure 4) because the gaps between higher quasienergies can be smaller than the gap between the near-degenerate EP quasienergies even quite close to the EP.

### 4.3   EPs with $\mathcal{PT}$ symmetry

If $N$ is odd and $L$ is even, one of the EPs appears on the real line with a negative value of $\lambda$. This is of some interest as for real $\lambda$ the system remains $\mathcal{PT}$-symmetric and can be endowed with unitary time evolution. This could serve as an interesting toy model of the passage of a $\mathcal{PT}$-symmetric system through an EP. If $N$ is even and $L$ is odd, or if $N$ is divisible by 4, there will also be an EP on the apparently $\mathcal{PT}$-antisymmetric line $\phi = 0.5$ described above. Examples of these negative-$\lambda$ EPs can be seen in Figures 5 and 6.

### 4.4   The case $k = n\pi$

Equation (5) always has the solution $k = n\pi$ where $k \in \mathbb{Z}$. This solution is not included in the analysis of Alcaraz et al. [12] or the work of Lieb et al. [25] on the XY model. It does not correspond to a quasienergy and does not contribute to the spectrum. However, there are values of $\lambda$ proportional to the $N$th roots of unity where a second root at $k = n\pi$ appears. At this point, one of the quasimomenta $k_j$ takes the value $\pi$, and changes from being real to complex. This was identified by Alcaraz et al. and earlier by Lieb et al., and occurs at the values

$$\lambda^N = \left(\frac{L}{L+1}\right)^{1/2}. \tag{23}$$

The corresponding quasienergy approaches 0 as $L \to \infty$ for $\lambda < 1$, implementing spontaneous breaking of the $\mathbb{Z}_N$ symmetry. Since the root is repeated, this point satisfies the quasienergy degeneracy condition Eq. (19). However, since one root is trivial and does not correspond to a quasienergy, this solution is in a sense spurious and does not produce a quasienergy degeneracy or EP. It is still of physical interest for the reason stated above.

### 4.5   The thermodynamic limit

The positions of the EPs can be determined in the thermodynamic limit $L \to \infty$ by examining the large-$L$ behaviour of Eq. (19). Following Alcaraz et al. [12] and Lieb et al. [25], $k_j$ can be expanded as

$$k_j = \frac{\pi j}{L} - \frac{\pi a}{L} + \mathcal{O}\left(\frac{1}{L^2}\right), \tag{24}$$

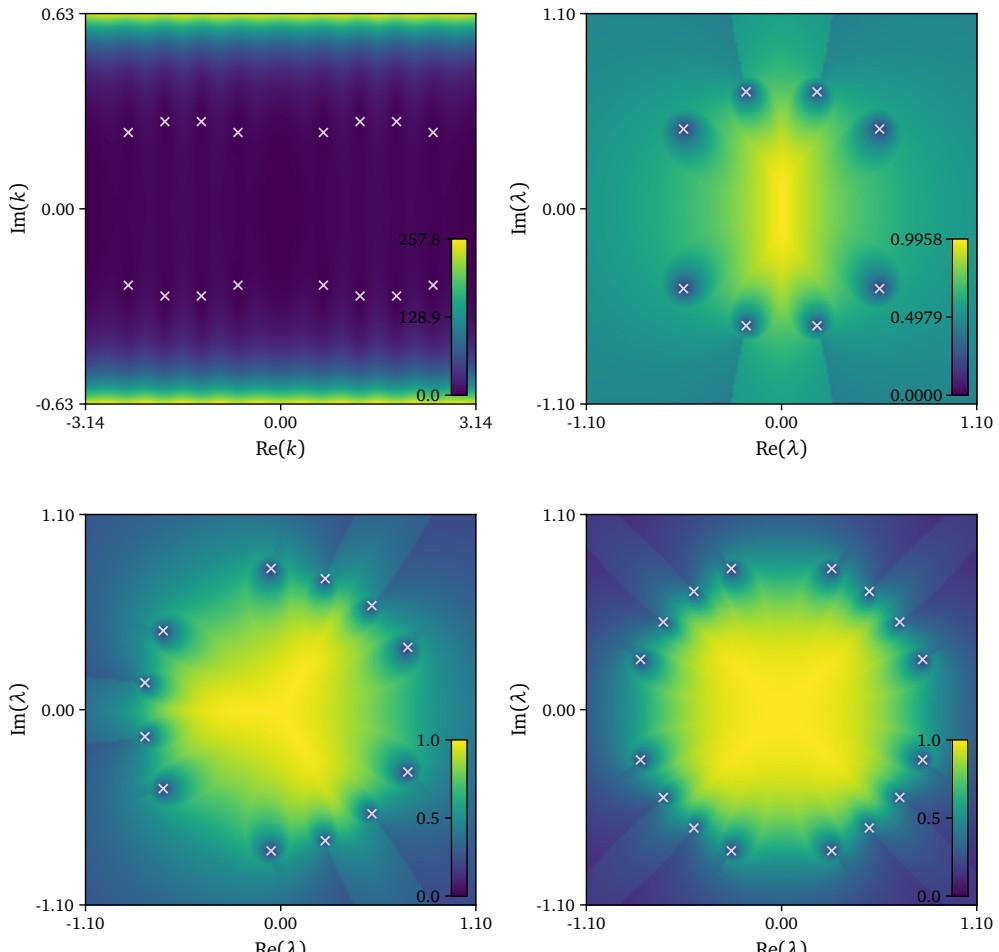

Figure 4: (Top left) Equation (19) evaluated for $L = 5$. The zeros are marked with white crosses. Also shown is the relative difference between the smallest and second-smallest quasienergies $\Delta\epsilon_{12}$ for $N = 2$ (top right), $N = 3$ (bottom left) and $N = 4$ (bottom right). The zeros from the left subfigure are transformed using Eq. (21) and marked with white crosses in each subfigure. They correspond to actual zeros of $\Delta\epsilon_{12}$ to numerical precision. There are additional zeros in (a) at $k = n\pi$, $n \in \mathbb{Z}$, but these do not correspond to a degeneracy as detailed in Section 4.4.

where $j \in \{1, \ldots, L\}$, and $a$ is determined by inserting the expansion into Eq. (17) (applying trigonometric sum formulae and discarding vanishing terms):

$$\cot(\pi a) = \frac{\lambda^{-N/2} + \cos(\pi j/L)}{\sin(\pi j/L)}. \tag{25}$$

Note that in Eq. (24), the term $\pi j/L$ is of order zero in $L$ since $j$ is of order $L$. The EPs are given by values of $\lambda_{EP}$ and $k_{EP}$ which satisfy both Eq. (17) and its derivative Eq. (18), which gives a second equation for $a$ at the EPs:

$$\tan(\pi a) = -\frac{\frac{L}{L+1}\lambda^{-N/2} + \cos(\pi j/L)}{\sin(\pi j/L)}. \tag{26}$$

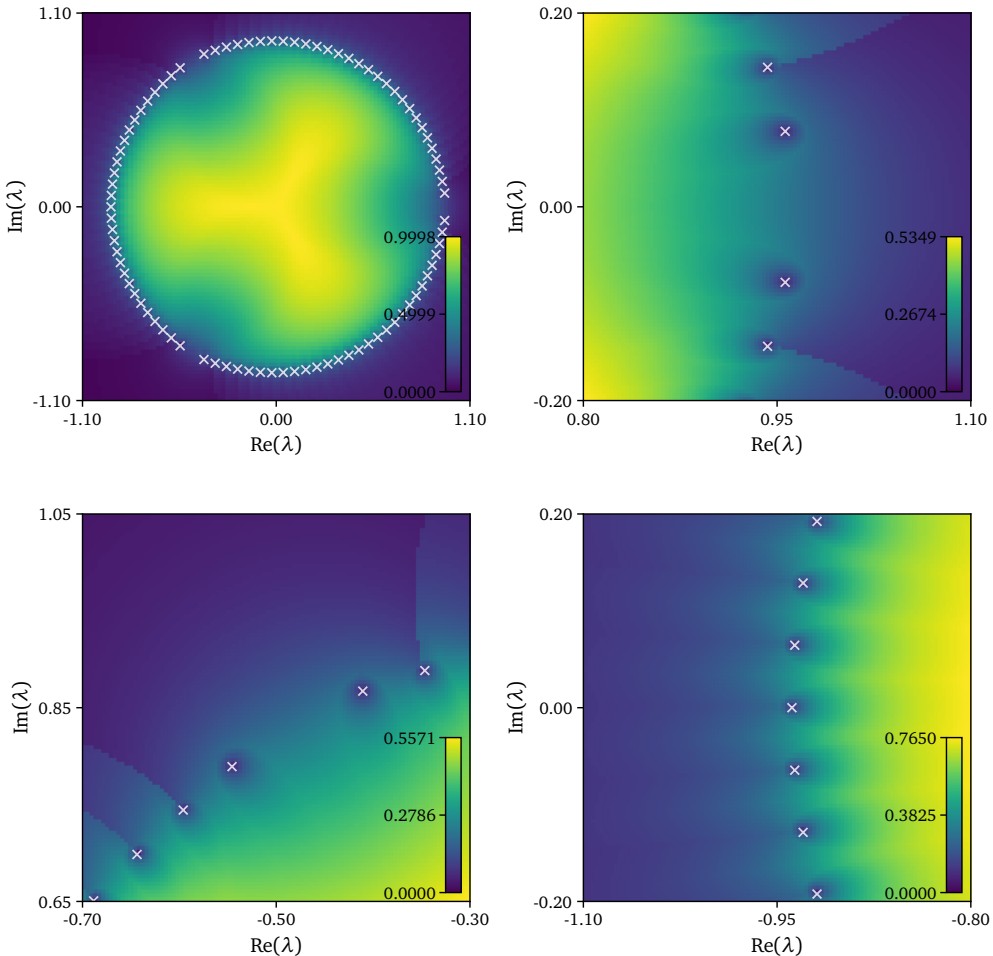

Figure 5: The absolute distance $\Delta\epsilon_{12}$ between the smallest two quasienergies for $L = 30$, $N = 3$. The exceptional points found by minimising Eq. (19) are marked with white crosses. Subfigures show different parameter ranges.

Combining the two and setting $L/(L+1) \approx 1$ gives

$$\frac{-\sin(\pi j/L)}{\cos(\pi j/L) + \lambda^{N/2}} = \frac{\cos(\pi j/L) + \lambda^{N/2}}{\sin(\pi j/L)}, \tag{27}$$

which by eliminating $a$ reduces to

$$0 = 1 + 2\lambda^{N/2}\cos\left(\frac{\pi j}{L}\right) + \lambda^N. \tag{28}$$

This has solutions

$$\lambda^{N/2} = -\cos\left(\frac{\pi j}{L}\right) \pm i\sin\left(\frac{\pi j}{L}\right), \tag{29}$$

which squares to (using double-angle formulae)

$$\lambda^N = \cos\left(\frac{2\pi j}{L}\right) \pm i\sin\left(\frac{2\pi j}{L}\right), \tag{30}$$

where $j \in \{1, \dots, L\}$ as defined above. These are precisely the $L$th roots of unity, with $j = L$ giving unity, and with each root appearing twice. This repetition results from the fact that for any $k$ defining an EP, or generally a quasienergy satisfying Eq. (17), the conjugate $k^*$ will give the same energy. Taking all solutions for $\lambda$, this gives all the $(NL)$th roots of unity. However, $N$ of these correspond to $k = 0$ and don't produce EPs, as described in Section 4.4. Thus the limiting case is the set of $(NL)$th roots of unity, with the $N$th roots of unity excluded, for a total of $N(L-1)$ EPs. Figure 5 shows numerical data for $L = 30$ which closely reflects this.

## 4.6 Other degeneracies of $H$

As per Section 2, quasienergy degeneracies are the only way that EPs of $H$ can appear. However $H$ may have other degeneracies, where the eigenvalues but not the eigenvectors become degenerate, and $H$ is still diagonalisable. In Fig. 6 the minimal relative distance between any two eigenvalues of $H$, $\Delta E$, is plotted, defined as

$$\Delta E = \min_{i,j} \left| \frac{E_i - E_j}{E_i + E_j} \right|, \tag{31}$$

where $E_i$ are the eigenvalues of $H$, and $i$ and $j$ range over all $N^L$ eigenvalues. This function has many zeros even for the small value of $L = 4$. Subfigures (b), (c), and (d) show various quantities along the real negative $\lambda$ line, which includes an EP at around $\lambda = -0.75$. Although there are many degeneracies of $H$ along this line (seen in subfigures (a) and (b)), only the degeneracy at the EP gives rise to orthogonal left and right eigenvectors, as seen in subfigure (d). Similar numerical tests have been performed for other small values of $L$ and $N$ and show the same behaviour, providing some numerical confirmation of the reasoning given in Section 2.

More generally, if the quasienergies are distinct then a degeneracy of $H$ will occur if some combination of the quasienergies sums to zero when multiplied by appropriate powers of $\omega$, as is clear from the form of the spectrum in Eq. (3), i.e.,

$$\omega^{k_1} \epsilon_{j_1} + \omega^{k_2} \epsilon_{j_2} + \cdots + \omega^{k_m} \epsilon_{j_m} = 0, \tag{32}$$

for some $m \leq L$. This is much more general than the quasienergy degeneracy condition $\epsilon_i - \epsilon_j = 0$. The simplest case of Eq. (32) is one involving only two quasienergies:

$$\omega^k \epsilon_i + \epsilon_j = 0, \tag{33}$$

for some $i$, $j$, and $k$, where the second power of $\omega$ has been divided out. It may be possible to evaluate simple conditions like Eq. (33) to find degeneracies analytically, however they will not satisfy the condition of repeated roots in Eq. (18) which easily allowed the determination of the EPs. This is a topic for further investigation, although these degeneracies do not have the same physical significance as the EPs.

## 5 Conclusion

The main result of this work is the identification of a series of $N(L-1)$ exceptional points in the complex-$\lambda$ plane of the free parafermion model, the locations of which are given by Eq. (19). As $L \to \infty$, these EPs approach the $(NL)$th roots of unity, with the $N$th roots excluded, i.e., they approach a uniform distribution on the unit circle. For complex values of $\lambda$, the $\mathcal{PT}$ symmetry of the model is destroyed. However, for odd $N$ and even $L$, one of the EPs exists at a negative real value of $\lambda$.

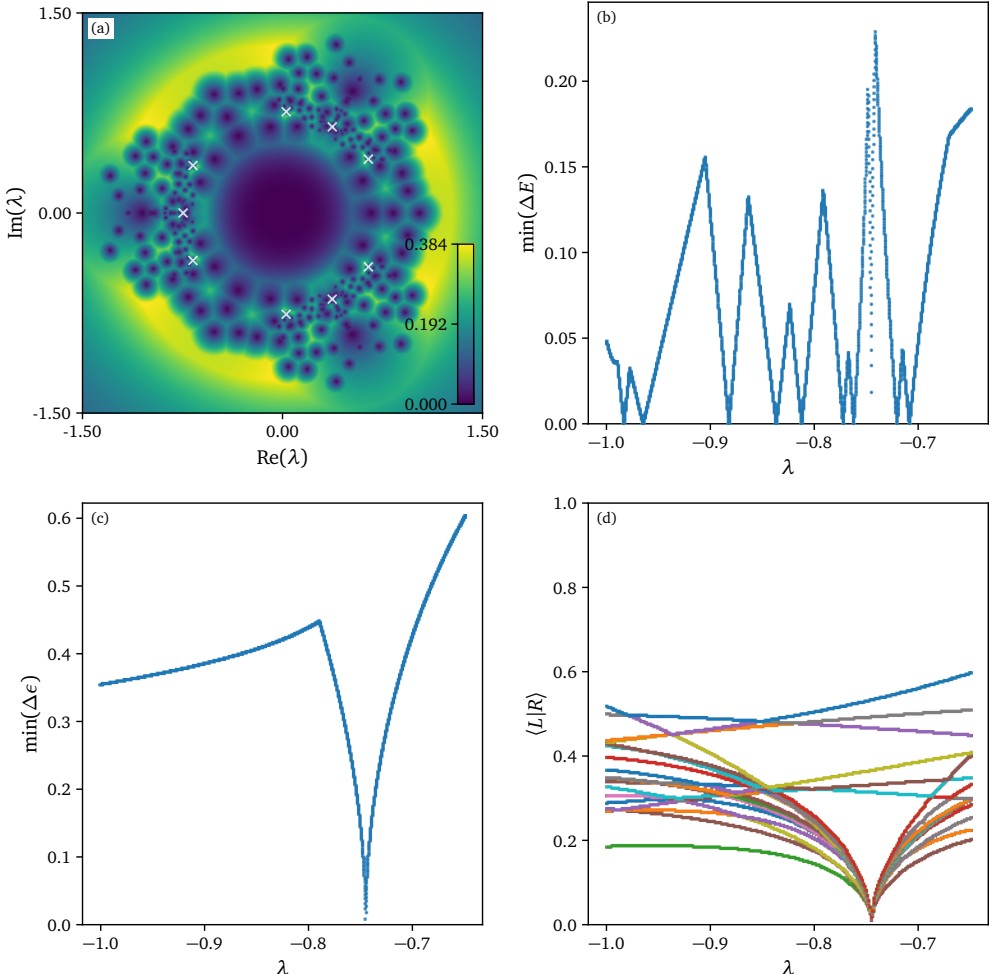

Figure 6: For $N = 3$, $L = 4$, (a) Smallest relative difference between any two eigenvalues of the Hamiltonian $\Delta E$. The EPs obtained from Eq. (21) are marked with white crosses. Although not clearly visible, these quasienergy EPs correspond to zeros of $\Delta E$. (b) $\Delta E$ for negative real values of $\lambda$, i.e., a line segment in subfigure a. (c) Smallest relative difference between the two smallest quasienergies $\Delta \epsilon_{12}$ for negative real $\lambda$. (d) Overlaps of the left and right eigenvectors of the Hamiltonian for negative real $\lambda$. Each colour shows a different left-right overlap.

In Fig. 2, the fidelity susceptibility of the ground state was observed to diverge with increasing system size for real positive $\lambda$ near the critical point $\lambda = 1$. This is characteristic of the system approaching an EP but could not be explained given real positive $\lambda$ since there was no way to achieve a quasienergy degeneracy. The large-$L$ limit of the complex-$\lambda$ EPs provides a clear mechanism for this behaviour, since as $L$ increases, infinitely many EPs approach the real axis at $\lambda = 1$.

Each of these EPs is a degeneracy of two of the quasienergies which define the free parafermion spectrum Eq. (3). This means that at each EP, any pair of energy levels that differ only by swapping the two degenerate quasienergies becomes degenerate. Thus each quasienergy degeneracy is in fact a set of $N^L - N^{L-1}$ 2-fold degeneracies, or $\frac{1}{2}(N^L - N^{L-1})$ coincident two-level EPs. Such coincident EPs have recently appeared in the literature and are termed *confluent* EPs [26, 27]. In particular, the passage of a system through such an EP has interesting physical properties. The free parafermion model may serve as a toy model for

such a passage, particularly in the $\mathcal{PT}$-symmetric case where unitary time evolution can be defined.

The EPs also exist in the familiar Ising spin chain, which is the limiting case $N = 2$ of the free parafermion model. They do not produce unusual behaviour on the real axis in this case, as the model is Hermitian for real $\lambda$. However the existence of these points appears to be unexplored in the literature, despite the fact that the Ising chain has been studied extensively. There are examples of non-Hermitian extensions of the Ising model such as a complex longitudinal field [28], but not to our knowledge of the direct extension of the transverse field $\lambda$ (or equivalent) to the complex plane. The behaviour of the EPs resembles Lee-Yang zeros [29], in that they appear on the unit circle, they appear as a consequence of making the model parameter complex, and they approach the critical point as $L \to \infty$.

# Acknowledgements

Some of the numerical calculations (DMRG and exact diagonalisation) were performed using the excellent TeNPy Library [30]. RAH thanks the staff of the National Computational Infrastructure facility at ANU for their generous advice and other assistance.

The authors thank the referee for raising a number of detailed points leading to clarifications and improvements in the final presentation.

**Funding information**   This work has been supported by the Australian Research Council through grant number DP210102243.

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
