# Peer review of "Exceptional Points in the Baxter-Fendley Free Parafermion Model"

_SciPost Physics, doi:SciPost Phys. 15, 016 (2023)_

## Round 1 · Referee Report · Anonymous (Referee 1) · 2023-3-9

Strengths

1- Study a set of interesting many-body model with two ingredients, parafermion degrees of fredonn and non hermitian .

2- Give a possible explanation of unusual effects reported on these quantum chains.

3- It can motivated further studies in the area of non hermitean quantum mechanics

Report

The authors study a set of Z(N) symmetric spin chains with a free-energy eigenspectrum. It is expressed as the composition of independent quasi energies. These hamiltonians introduced by Baxter and solved by Fendley have exclusion effects in the spectral combination of the quasi-energies, that depend on N, and they are called free parafermion models. For N>2 they are non hermitean, and may also show unusual physics as compared with usual hermitian quantum mechanics.

In a previous publication the authors observe an usual behavior of the correlation function, for small chains of the N=3 model. In this paper the authors give a possible explanation for this behavior. For that, they extend the usual real value coupling constant defining the quantum chain to complex values, as is usual for the study of Yang-Lee zeros of partition function, on this way the quasi-energies that are all disting for real coupling values can become equal at special values of complex coupling constant parameters.

At these points where the eigenspectrum becomes globally degenerated they verify numerically for small lattice sizes, but they also verified that these degenerated states are parallel, and they call this point as an Exceptional Point. The authors mention that from general studies in non hermitian quantum mechanics unusual effects happens in these cases.

They give numerical evidence that in the thermodynamic limit , for the N=3 case, exceptional points that happens for imaginary couplings can reach a real value for the coupling constant, possibly the critical point.

In my opinion the paper should be published since it exemplifies an exactly many-body integrable model with the unusual effects predicted in non-hermitian quantum mechanics, and my be useful as a toy model in this area of research. I have however some remarks.

1) At these EP, the coalescence of the eigenvector is not just the appearance of a standard Jordan cell. If this would be the case, I think the authors should mention.

2) Equation (25) has an important misprint

---

## Round 1 · Referee Report · Anonymous (Referee 2) · 2023-3-14

Strengths

  • draws attention to a model still little explored in the literature

  • EPs are easily determined

Weaknesses

see report

Report

The authors consider a simple solvable Z(N) Hamiltonian introduced by Baxter in the late 1980s [1]. This is one of the simplest possible generalizations of the Ising chain in a transverse field. Recently, the model was solved in terms of free parafermions by Fendley [3] and since then there has been renewed interest in its solution, although the model is non-Hermitian for N>2.

In the present manuscript, the authors' main result is the determination of the so-called exceptional points (EP). These points occur when eigenvalues degenerate and the associated eigenvectors become parallel (coalesce). They are relevant for non-Hermitian Physics. For Baxter's model the EPs are given by equations (13) and (14). The transverse field is general complex at an EP. The coalescence of the eigenvectors is verified numerically.

I think the results in the paper are nice and can be published in SciPost, provided that the following below are addressed.

Requested changes

1- The determination of EPs is quite simple, as an analytical form of the quasienergies - eqs.(4,5) - is known. Therefore, I think the authors could make an effort to understand the coalescence of the eigenvectors analytically/algebraically. This should be feasible at least for for the case $N=2$.

2- The authors should be precise about which correlation functions are divergent for the Baxter-Fendley model, and how this divergence is explained by the exceptional points. Although this seems to be the main motivation of the paper, little is said or done about it.

Other minor comments:

3- Verify Fig. 2. For example, the top right panel ($\phi=0$) seems inconsistent with Eq. (4).

4- Verify Fig. 3. The pictures seems to be associated with L=5 not L=4.

5- The quantity $\Delta\epsilon_{01}$ in Fig. 3 should be clearly defined, maybe in the main text. Since the quasienergies are complex, it is not clear how the authors defined smallest and second-smallest quasienergies. Same for $\Delta E_{01}$ in Fig. 6.

6- I think Fig. 4 should be improved. It is not very clear what is plotted. Since this picture is suposed to confirm an important result of the paper, I think readers would benefit from more clear and well defined labels.

7- It seems that there is a misprint in the definition of $\lambda$ in Fig. 7 , as it does not give $\lambda=1$ when $\gamma=1/2$.

8- In the Conclusion, the authors mention that the number of EPs is $NL$. Clarify if this number includes the trivial EPs associated $k=0$ and $k=\pi$ . Maybe this number should be mentioned in Sec. 3 (see also remark 4-).

---

## Round 2 · Referee Report · Anonymous · 2023-4-21

Report
I thank the authors for clarifiyng the scale issue and the ordering of the quasienergies.
I now agree with the cases $\phi=0$ and $\phi=0.1$ in Fig. 3, but unfortunately I think $\phi=0.5$ is not correct.
As in the previous version by the authors, it seems that the quasienergies do appear in complex pairs, even without a further rotation. Here are the numerical values that I found solving eq.(4,5) for $|\lambda|=1,\phi=0.5$:
k-values: $\{-2.4553+0.0826589 i,-1.82173+0.202561 i,-1.31986+0.202561 i,-0.686291+0.0826589 i\}$
quasienergy values: $\{0.634754\, +0.584635 i,0.634754\, -0.584635 i,0.990265\, +0.601812 i,0.990265\, -0.601812 i\}$
So I ask the authors to please check this point one more time.

---

## Round 2 · Author Response

We would like to thank the reviewer for their additional comments. Below are our responses to the referee's new comments.
- The reviewer is right: Figure 3 had an incorrect scaling function applied. This has been corrected and we agree with the reviewer's numerical values. We have verified that the values shown in Figure 3 can be obtained directly from Equations 4 and 5. Following this, we also decided to remove the global rotation of the Hamiltonian given in Section 4.1, as it is unnecessary to most of the paper and has no effect on the EPs. It is still discussed in 4.1 along with the PT-antisymmetric phi=0.5 case.
- We have added a definition of Delta(epsilon) and explained why this quantity is used over another possible choice.

---

## Round 2 · List of Changes

+ The data in Figure 3 had been rescaled incorrectly and has been fixed.
+ The redefinition of the Hamiltonian to include a global rotation in Section 4 has been removed. These rotations are still discussed in Section 4.1.
+ The removal of the global changes the values shown in Figure 3, which have been recalculated (they are rotated by 2pi/6). Figure 3 now corresponds directly to Equations 4 and 5.
+ The removal of the global rotation does not affect the other figures as they only depend on differences of (quasi)energies.
+ A definition of Delta(epsilon) is now given in Equation 22 along with some discussion of why we use this quantity. This quantity has also been relabeled with subscript 12 instead of 01 as the quasienergies are in fact labelled from 1, not 0.

---

## Round 3 · Referee Report · Anonymous (Referee 6) · 2023-4-24

Report

I thank the authors for explaining in detail the conventions used. In my opinion the paper is ready to be published in SciPost.

---

## Round 3 · Author Response

Dear editor and referees,

We apologise for the confusion on this point, and we appreciate your patience.

The referee is indeed right, if a convention is applied to make the quasienergies have the smallest complex argument possible, or equivalently to have arguments in the range (-pi/N, pi/N]. There are N equivalent choices for each quasienergy corresponding to the N roots of Equation 4. We think this is a good convention to apply because it reveals that the quasienergies are conjugate pairs for phi=0.5 as the referee observed. The quasienergies previously shown in Figure 3 also satisfy Equations 4 and 5 but we have switched to this convention as it's a clear improvement.

---

## Round 3 · List of Changes

+ Figure 3 has been changed to use the convention that the quasienergies have arguments in the range (-pi/N, pi/N].
+ This convention is explained in the text in Section 4.
+ The discussion of the antisymmetric point is corrected in light of this convention.
+ Applying this convention does not affect any of the other material in the paper.
+ An acknowlegement of the referees' contributions has been added.

---

## Editorial Decision

published